# Leveraging Human Revisions for Improving Text-to-Layout Models

## Abstract

Learning from human feedback has shown success in aligning large, pretrained models with human values. However, prior works have mostly focused on using high-level labels, such as preferences between pairs of model outputs. On the other hand, many domains could benefit from more involved, detailed feedback, such as revisions, explanations, and reasoning of human users. Our work proposes using nuanced feedback through the form of human revisions for stronger alignment. In this paper, we ask expert designers to fix layouts generated from a generative layout model that is pretrained on a large-scale dataset of mobile screens[1]. Then, we train a reward model based on how human designers revise these generated layouts. With the learned reward model, we optimize our model with reinforcement learning from human feedback (RLHF). Our method, Revision-Aware Reward Models (RARE), allows a generative text-to-layout model to produce more modern, designer-aligned layouts, showing the potential for utilizing human revisions and stronger forms of feedback in improving generative models.

## 1 Introduction

Large, pretrained models have shown impressive results in many domains, including natural language and text-to-image generation. However, because these models are typically trained on large-scale, unfiltered data, they may not be aligned with human values. To ensure positive usage of these models. it is important to address the issue of unsafe, inaccurate, or outdated generations.

A developing area of study attempts to address the misalignment problem by learning from human feedback. Lee et al. (2023) and Liu et al. (2023) use binary human preferences to align text-to-image diffusion models and large language models respectively. Scheurer et al. (2023) augments preference feedback with language feedback to finetune large language models. While these methods have improved alignment, it still remains unclear what type of feedback to collect, and how to best utilize it.

Prior works have primarily focused on using high-level labels of human feedback, e.g., preferences between pairs of model outputs such as images or languages. However, in the real world, we would learn better when relying on detailed corrections, explanations, and reasoning. We hypothesize that learning from human revisions is more effective for a model to adapt to produce human preferred results. Compared to preferences or language feedback, which is previously used for human feedback, revisions, as a type of feedback, not only indicate human preferences on the end results but also provide nuances in how to align model outputs with human expectations.

In this paper, we investigate this approach in the domain of text-to-layout generation in which a model is trained to generate a layout given a text input. The task has gained increasing interest from the field as it can significantly reduce the effort of designers in creating a layout. To do so, we first ask professional designers to improve layouts generated from PLay (Cheng et al., 2023), a generative layout model that is trained on a large-scale dataset of mobile screens—the dataset reflects earlier generations of Android UIs. Designers perform these revisions in Figma, a popular design tool, and our plugin records detailed, step-by-step edits that designers perform in revision a layout towards their satisfaction. Based on these revision sequences, we train a reward model, which is then used to optimize the generative model, using reinforcement learning from human feedback (RLHF).

---

[1]Our design revision dataset will be released on the GitHub.

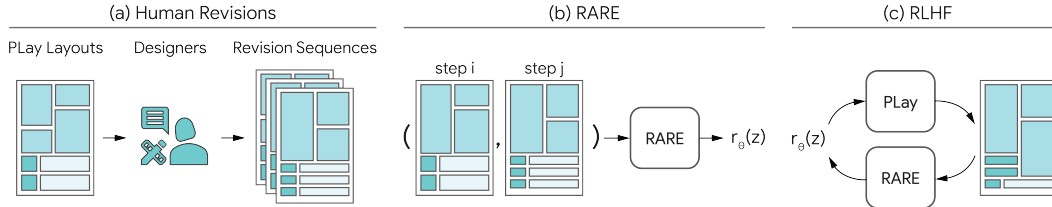

Figure 1: **RARE Method Overview** Our method consists of three parts: (a) Collection human revision sequences, (b) training our reward model on sequence data, and (c) using the reward model in an RLHF framework.

Based on our experiments, our method, Revision-Aware Reward Models (RARE), significantly outperforms the baseline method that directly fine-tunes a generative model with a supervised approach. Our experiments also showed that reward signals that are designed based on revision sequences lead to more desirable outcomes than using preferences alone. By analyzing the outputs acquired by our method, we found our method leads to more modern, designer-aligned layouts, even though the base model was trained on a dataset with old-fashioned Android designs. These show the potential for utilizing human revisions and stronger forms of feedback in improving text-to-layout models.

**Contributions** We highlight our contributions as follows:

- We collect a high-quality, detailed dataset from experienced designers, producing over 2,000 sequences of revisions from the generated layout to the human-revised layout.

- We propose a method, RARE, for learning from human revisions of layout design, and examine three variants of RARE, including Keystroke, Chamfer, and Preference.

- Our experiments and analysis show that reward models that use human improvement-specific data can be used in RLHF to effectively finetune the base text-to-layout model for qualitative and quantitative improvements.

## 2    RELATED WORK

Human priors have been used to guide language model alignment (Liu et al., 2023; Ouyang et al., 2022), text-to-image models (Lee et al., 2023; Fan et al., 2023; Black et al., 2023), and behaviors (Hejna & Sadigh, 2022; Lee et al., 2021). These human priors can be represented through a variety of manners, including (1) Extracting rewards from pretrained foundation models like Stable Diffusion (Rombach et al., 2022b; Jain et al., 2022) or R3M (Nair et al., 2022; Adeniji et al., 2023); (2) Curating high quality, human-aligned datasets for finetuning (Ouyang et al., 2022); (3) Explicitly learning reward functions from human feedback (Xu et al., 2023; Lee et al., 2023; Bai et al., 2022; Stiennon et al., 2022). Our work utilizes human feedback for alignment.

**Types of Human Feedback** The most informative types of human feedback is still an open area of research. Prior works in generative models have primarily used binary human preferences or comparisons (Bai et al., 2022; Liu et al., 2023; Lee et al., 2023), scalar ratings (Stiennon et al., 2022). To our knowledge, there are no works that actively leverage human revisions to the generated outputs of generative models, which is a stronger and more involved form of human feedback that we propose.

Notably, correctional feedback has been used in robotics applications (Li et al., 2021; Losey et al., 2021; Bajcsy et al., 2018; Ross et al., 2011). However, these robotics-focused works focus on improving a trajectory or a sequence of actions, which is a multi-step bandit problem. Our work focuses on improving generative samples, which is a one-step bandit problem.

**Learning from Human Feedback** Given human-annotated data, a popular approach is reinforcement learning from human feedback (RLHF). RLHF consists of a two-stage process: (1) Training a reward model on human feedback and (2) Optimizing a reinforcement learning objective. This has shown success in language modelling (Casper et al., 2023), text-to-image models (Lee et al., 2023; Black et al., 2023), and more.

Reinforcement learning-free methods can also learn from human feedback. A traditional method is supervised finetuning on the curated dataset (Ouyang et al., 2022). Recent papers have proposed new supervised objectives, such as Chain of Hindsight (Liu et al., 2023) and Direct Policy Optimization (Rafailov et al., 2023), to align large language models with human feedback.

**Layout Generation** For generation tasks such as layout design, layouts are saved in a vector graphic format, so that designers can easily edit and use them for downstream purposes. This modality is amenable for collecting human revisions. Recent studies on layout generation use sequential models like Transformers (Vaswani et al., 2017) to output the layout elements as sequences (Gupta et al., 2021; Arroyo et al., 2021; Kong et al., 2022; Kikuchi et al., 2021). LayoutDM and PLay (Inoue et al., 2023; Cheng et al., 2023) show results in conditional layout generation. We choose PLay as our backbone model based on its flexibility to inject different types of conditions using latent diffusion(Rombach et al., 2022a) and classifier-free guidance (Ho & Salimans, 2022).

# 3 BACKGROUND

## 3.1 DIFFUSION MODELS

Diffusion models are a popular class of generative models that learns a denoising processing from a known Gaussian prior to the data distribution. During training, the diffusion model learns to reverse a single noising step, which reduces to the following training objective:

$$\mathcal{L}_{DDPM}(\phi, \mathbf{z}) = \mathbb{E}_{t,\epsilon}[w(t)||\epsilon_\phi(\alpha_t \mathbf{z} + \sigma_t \epsilon) - \epsilon||^2] \tag{1}$$

$\phi$ are the learned parameters, $\mathbf{z}$ is a real data sample, $\epsilon$ is the added noise, $t$ is a scalar time step that schedules noise values $\alpha_t, \sigma_t$.

To sample from the diffusion model, the diffusion model iteratively denoises the initial sample $\mathbf{z}_T$ from the known prior $N(0, 1)$, where $T$ is the number of denoising steps, and $x_0$ is the final sampled data point. Denoising steps are modelled as Gaussian distributions:

$$p_\phi(\mathbf{z}_{t-1}|\mathbf{z}_t) = N(\mathbf{z}_{t-1}|\epsilon_\phi(\mathbf{z}_t, t), \sigma_t^2 I) \tag{2}$$

Diffusion models can be conditioned on additional context $\mathbf{c}$, whether it is text (Rombach et al., 2022b), guidelines (Cheng et al., 2023), or more. Furthermore, to reduce computational costs, diffusion models are often trained in latent space.

## 3.2 GENERATIVE LAYOUT MODELS

Layout designs are used by engineers and designers to produce vectorized arrangements and models. There are several commonly used datasets for layout modeling, including PublayNet (Zhong et al., 2019), CLAY (Li et al., 2021), and RICO-Semantic (Sunkara et al., 2022). In this paper, we focus on UI layouts, which consist of a collection of UI elements. Each UI element has a class type such as Button or Checkbox, and a bounding box that specifies the position and size of the element. In particular, we look at the task of generating UI Layouts based on text input, which is useful for assisting UI designers to create design mockups. Previously, PLay (Cheng et al., 2023), a conditional latent diffusion model, generates layouts based on a given set of guidelines. In this work, we train PLay to generate a UI layout based on a text input, such as "a login screen". We condition on text prompts instead of guidelines because text inputs give designers more context for the layout design, aiding them in generating meaningful revisions. To obtain the paired text prompts for CLAY, we define a screen summary task and feed the view hierarchy of each layout to a pre-trained large language models, PaLM (Chowdhery et al., 2022).

# 4 METHOD

Our objective is to better align a generative layout model such as PLay (Cheng et al., 2023) with human preferences. First, we collect a dataset of how human improvements, where experienced

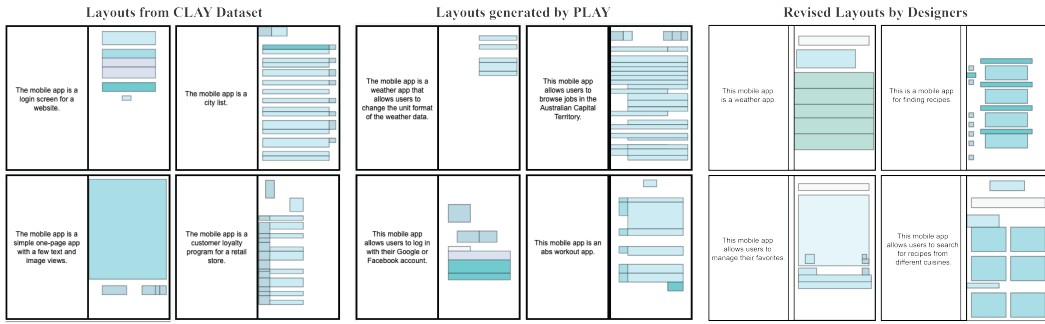

Figure 2: **Layouts Visualizations** We use CLAY (left) as our pretraining dataset. PLay (middle) is a generative, text-conditioned layout model. We ask designers to edit layouts generated by PLay, leading to more modern, cohesive layouts (right).

designers are asked to improve UI layouts generated by human designers. Next, we learn a reward model from the human revision dataset. Finally, we use reinforcement learning from human feedback (RLHF) to improve our base model, using various designs of reward signals.

## 4.1 HUMAN REVISION DATA COLLECTION

**Figma Plugin** To facilitate data collection, we use Figma[2], a design tool commonly used by designers. Our Figma plugin visualizes mock-ups generated by PLay and its corresponding text condition. The plugin uses text boxes and elements from Material 3 design library[3] to represent the various classes. Our plugin records all the operations performed by the designer as well as corresponding layout states in the process of revising a layout.

**Human Revisions** We recruit 4 professional designers to revise layouts generated from PLay. Designers are asked to modify the layout to be more aesthetic and coherent. For instance, we expect designers to fix misaligned elements or change the format of the page according to the text description.

Designers are able to move, scale, and change classes of elements. Designers may add or remove elements as necessary. We conducted the study asynchronously without time restrictions. After the designer completes the task, we save the sequence of edits and the final layout.

Our final dataset $\mathcal{D}_{\text{human}}$ consists of revision sequences $\{\{b_i^j, l_i^j\}_{j=0}^M, d_i\}_{i=0}^N$. At each revision step $i$, $b_i^j$ and $l_i^j$ are the bounding boxes and class labels for the $M$ elements in layout. $d_i$ is the time duration starting from the $i$th revision to completing the revision sequence, which reflects how much effort it takes the designer to transform the layout at the step into its final form. Finally, $N$ is the number of total revisions the designer makes.

**Sample Revision Sequence**

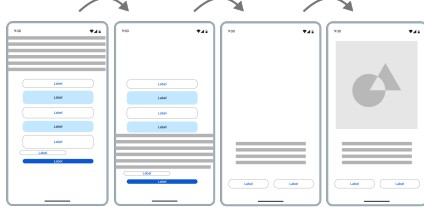

"Screen displaying user information in a dating application"

Figure 3: **Figma Plugin** Our Figma plugin renders a PLay layout with the corresponding text description. Designers are asked to revise the layout by adding, modifying, and deleting elements.

## 4.2 REWARD LEARNING

To maximally leverage the revision data, we propose **Revision-Aware Reward Models (RARE)**. RARE predicts the amount of effort needed to improve the layout. Unlike preference or binary ratings, RARE produces a scalar value roughly corresponding to the quality of the data point: low revision effort implies a strong layout, and high revision effort implies a poor layout.

---

[2] https://www.figma.com/
[3] https://m3.material.io/

In practice, we define the effort as the amount of time needed for the designer to complete the revision of the layout. At each revision step of our revision sequence, we have time $d$, the amount of time needed to complete revision of the layout, and the layout, which we encode to get latent vector $z$, obtained by using PLay's first stage latent autoencoder. From this, we construct a supervised learning problem for our reward model $r_\theta$:

$$\mathcal{L}_{\text{RARE}} = \mathbb{E}_{(\mathbf{z},t)\sim\mathcal{D}_{\text{human}}}[(r_\theta(z) - d)^2] \tag{3}$$

Because our revision dataset is limited, we first pretrain our reward model on the CLAY dataset. We procedurally perturb the layout by randomly revising, adding and dropping elements from the original sequences. For the revised elements, we randomly resize them to between $0.5$ to $2.0$ times of their original widths and heights, and we randomly move the elements uniformly between one width and height lower than its original position, and one width and height higher than its original position, bounded by the edges of the layouts. We randomly drop and add generated elements up to $1.5$ times the original numbers of elements in the original sequences. Then, for each dropped element, we assign a cost of $1$ time step to it, and $2$ time step for each revised element, and $3$ time step for each added element. We normalize procedurally generated time durations and dataset time durations to minimize discrepancies. After pre-training the reward model on this synthetic dataset, we can efficiently finetune our reward model from $\mathcal{D}_{\text{human}}$.

### 4.3 Reinforcement Learning from Human Feedback

Finally, to align our model with human feedback, we use RLHF. Following Black et al. (2023), we treat the learned denoising process as a Markov Decision Process, where the environment is represented as the following.

$$\mathbf{s}_t \triangleq (\mathbf{c}, t, \mathbf{z}_t) \quad \pi(t|\mathbf{s}_t) \triangleq p_\phi(\mathbf{z}_{t-1}|\mathbf{z}_t, \mathbf{c}) \quad P(\mathbf{s}_{t+1}|\mathbf{s}_t, t) \triangleq (\delta_{\mathbf{c}}, \delta_{t-1}, \delta_{\mathbf{z}_{t-1}})$$

$$a_t \triangleq \mathbf{z}_{t-1} \quad \rho_0 \triangleq (p(\mathbf{c}), \delta_T, \mathcal{N}(0, I)) \quad \mathcal{R}(s_t, a_t) \triangleq \begin{cases} r_\theta(x_0, \mathbf{c}) & \text{if } t = 0 \\ 0 & \text{otherwise} \end{cases} \tag{4}$$

$c$ is the conditioned text prompt, $\pi$ is our policy, $\rho_0$ is the initial state distribution, $\delta$ is the Dirac delta distribution, $T$ is the number of DDPM sampling steps, and $r_\theta(x_0, \mathbf{c})$ is our reward RARE.

We optimize for the reward by using DDPO with importance sampling. The DDPO algorithm is based on Proximal Policy Optimization (Schulman et al., 2017), which clips importance sampling weights to constraint update steps.

## 5 Experiments

### 5.1 Experimental Setup

**Base Model** We use a variation of the PLay model, which is conditioned on text input instead of grid-based guidelines in the original paper. We pretrain the text-conditioned PLay, as our base model, on CLAY (Li et al., 2022), a public dataset, and corresponding text labels from CLAY screens are generated by PaLM (Chowdhery et al., 2022), an LLM. The CLAY dataset is derived from RICO (Deka et al., 2017), a popular mobile corpus that consists of UI screens of early Android verions. The hyperparameters for training text-conditioned PLay are included in the Appendix.

**Dataset** We use a dataset of 836 revised UI Layouts from designers. The average revision sequence length is 88.9, leading to a total of 8,694 unique layout revision sequences. We split the dataset into 645 train examples and 191 evaluation examples. Statistics on edit time and types of revisions are included in Figure 7, and distribution shifts in element classes are presented in Figure 5.

**Baselines** We evaluate the following methods:

1. Supervised Finetuning (SFT): We directly finetune PLay on the final revisions.
2. Preference Reward + RLHF: We train a preference reward model on pairs extracted from the revision sequences. We assume that the final revision is the most optimal, so all other intermediate revisions are considered negatives.

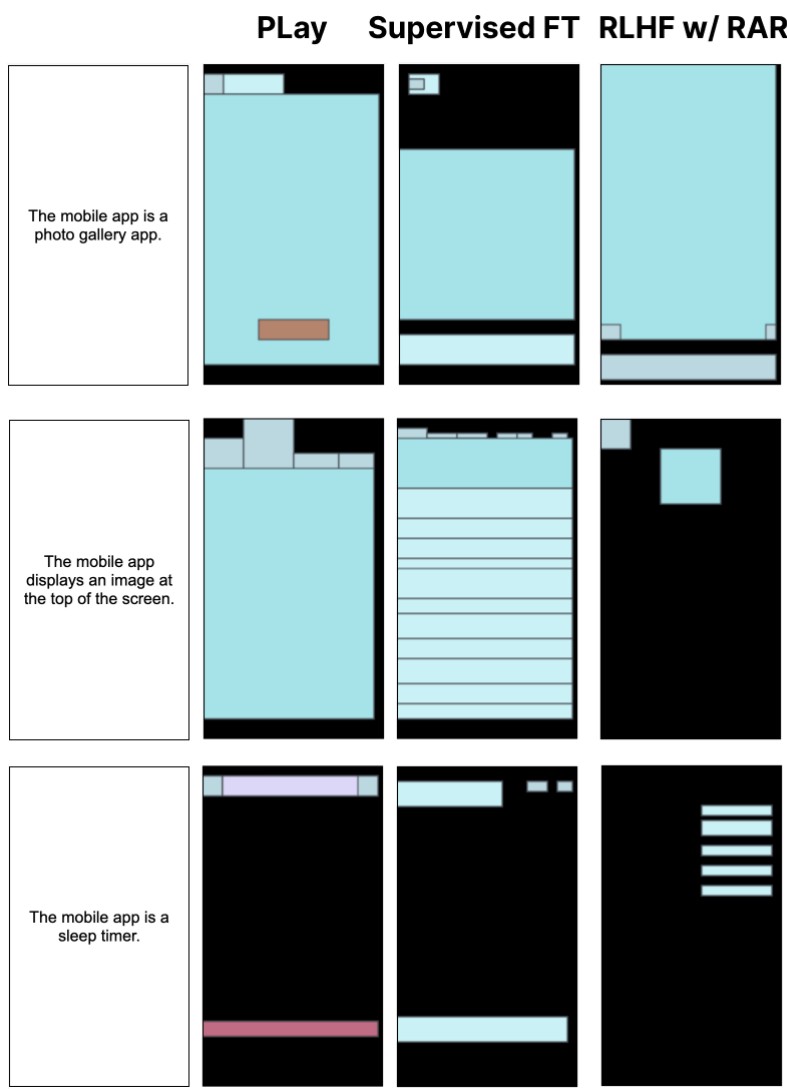

Figure 4: **Result Comparison** We compare layouts generated by a PLay model, a supervised fine-tuned model, and a model trained with RLHF with RARE. In these examples, RLHF w/ RARE produces the most cohesive and aligned layouts.

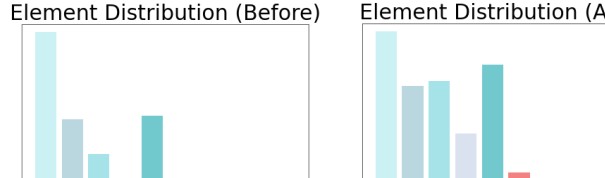

Figure 5: **Element Distribution** The element distribution from PLay samples (left) becomes more diverse after designer revisions (right). Designers add more elements (mean number of elements increases from 11.02 to 13.05 after revisions), particularly images and labels.

|  |  | FID Score ($\downarrow$) | DocSim ($\uparrow$) |
|---|---|---|---|
| Dataset | CLAY Dataset | $73.4 \pm 4.5$ | $0.14 \pm 0.01$ |
|  | Revision Dataset | $63.1 \pm 3.0$ | $0.16 \pm 0.01$ |
|  | PLay Samples | $76.6 \pm 0.3$ | $0.31 \pm 0.002$ |
| Finetuning | Supervised FT | $68.9 \pm 0.6$ | $0.26 \pm 0.01$ |
| RLHF | RARE Keystroke | $70.1 \pm 0.8$ | $0.27 \pm 0.03$ |
|  | RARE Preference | $72.4 \pm 2.6$ | $0.31 \pm 0.02$ |
|  | RARE Chamfer | $68.8 \pm 0.5$ | $0.28 \pm 0.01$ |

Table 1: We evaluate the FID Score and DocSim with a held-out batch of human revisions. We find that RARE Keystroke exceeds Supervisied Finetuning in DocSim and matches Supervised Finetuning in FID score.

3. RARE Chamfer Distance + RLHF: We learn the Chamfer Distance from an intermediate layout to the final layout and utilize the negative distance as the reward. The Chamfer distance is a geometric distance that is revision-aware (RARE).

4. RARE Keystroke + RLHF: We learn the time difference from an intermediate layout to the final layout and utilize the negative predicted time as the reward. This time-based metric is revision-aware (RARE).

All rewards are normalized for more efficient RL training. For our RLHF methods, because our object is to lightly finetune our model, we use a standard DDPM sampler, but we only optimize the DDPO objective on the last 10 timesteps of the denoising diffusion model. We find that further optimization for early steps of the denoising process leads to mode collapse and reward exploitation. For additional hyperparameters, please refer to the Appendix.

## 5.2 QUANTITATIVE RESULTS

To evaluate the alignment of our finetuned model with the human revision dataset, we report the FID scores with the final revisions in Table 1. We also report the DocSim score Patil et al. (2019), a popular measure of similarity across documents. Notably, the FID score between the CLAY dataset and the human-revised layouts is high, suggesting that there is a large distribution shift between the types of layotus collected in CLay and those that are preferred by and created by designers.

In our results, we find that RARE Chamfer the lowest FID scores. This supports our hypothesis that revision sequences are informative and effective for specifying human preferences.

Supervised Finetuning is less effective for finetuning, with a lower DocSim score. The FID score is much higher than RARE RLHF, and we hypothesize that this may be because our dataset is rather limited, and finetuning the entire model is ineffective for alignment.

Finally, we find that using the Chamfer Distance is far less effective than RARE Keystroke. This result is interesting, because both the Chamfer Distance and Keystroke seek to quantify the amount of human effort needed to fully revise the layout. We hypothesize that this may be because the Chamfer Distance is much harder to learn, especially on a limited dataset.

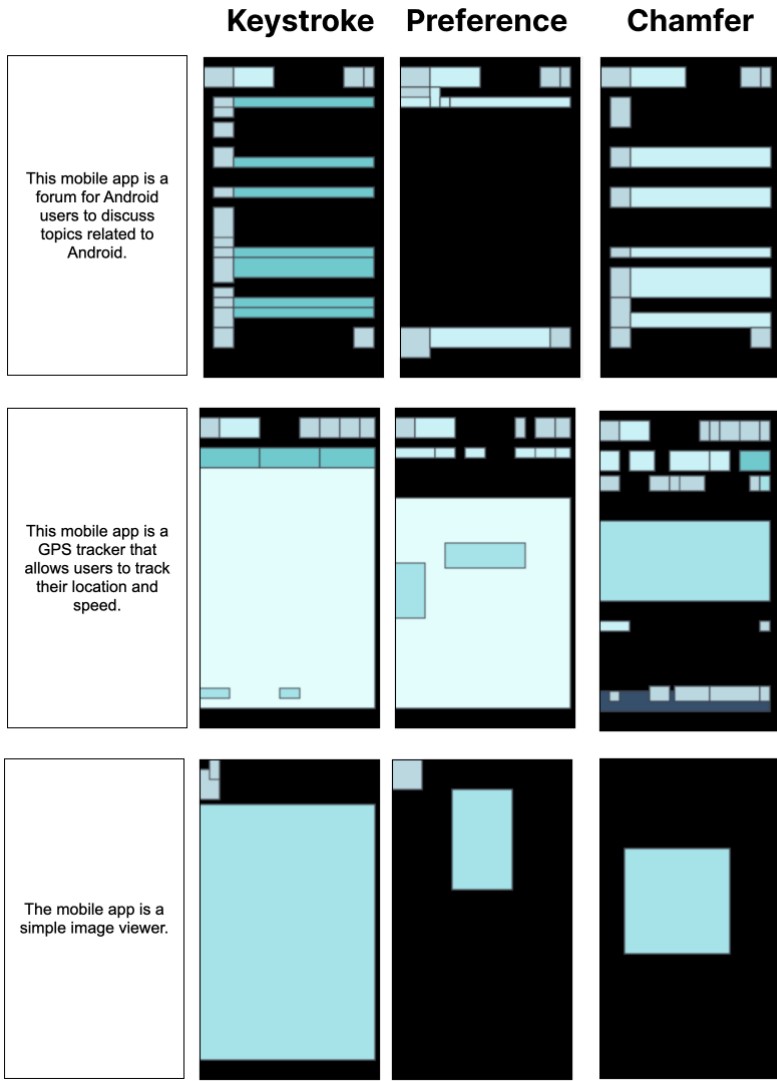

Figure 6: **RLHF with Different Reward Models** We compare the effect of using different reward models. We find that RARE Keystroke and RARE Chamfer lead to more consistent and coherent samples. For instance, in the first row, the RARE Chamfer and RARE Keystroke samples may resemble forum discussions more. In the second example, RARE Chamfer and Keystroke samples are well-aligned. In the last example, we find that RARE Keystroke generates a large image, which is typically unseen in the pretraining data, suggesting successful alignment with human revisions.

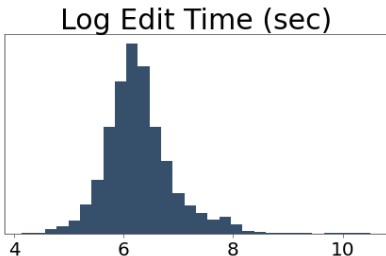 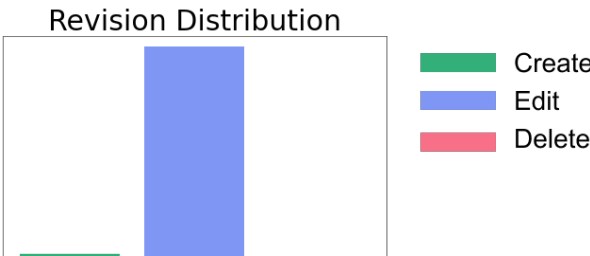

Figure 7: **Revision Statistics** We plot the distribution of natural log of the edit times (left) and the distribution of types of revisions (right). The median time is 503.4 seconds for a designer to complete a full revision. The overall distribution is skewed by a couple of outliers that took an extremely long time, and when taking the natural log of the edit time, it resembles a normal distribution.

### 5.3 QUALITATIVE COMPARISONS

We show qualitative examples of how RARE may better align layouts with user preferences. In Figure 4, we compare layouts from PLay (the base model), PLay + Supervised Finetuning, and PLay + RLHF w/ RARE. We find that samples from the Supervised FT model differ greatly from the base model and are inconsistent in quality. However, RLHF w/ RARE stays close to the base model samples, slightly improving it based on the learned design principles. This supports our hypothesis that RARE provides informative feedback and can guide the layouts to be more human-preferred.

Next, we compare how different reward models trained on the revision sequences affect the quality of the generated outputs. In Figure 6, we compare RARE, preference modelling, and Chamfer distance. Although RLHF with preference reward models achieves a similar FID score to RARE, qualitatively, we notice qualitative differences in alignment and overall layout, described further in detail in Figure 6.

### 5.4 DATASET ANALYSIS

We provide additional insight to how designers are revising layouts in Figures 7. Designers take a median time of 503.4 seconds per full revision. The mean number of elements before is 10.4 (standard deviation 7.9), and the mean number of elements after revision is 15.8 (standard deviation 9.0).

In addition, we plot the element class distributions from before and after revisions in Figure 5. From the revision data and designer feedback, we find that the base PLay generations are not optimal for most use cases. The shift in element class distributions and the amount of time spent revising the layouts suggests that many elements are misplaced or of the wrong class. This may be partially due to the fact that the base model PLay was trained on an older dataset of UI screens, motivating the need for human alignment.

## 6 CONCLUSIONS

In this work, we present a method for leveraging detailed human feedback through the form of revision sequences. We ask designers to revise layouts generated a text-conditioned generative layout model. Using the revision data, our method, RARE, predicts the amount of time between an intermediate and the final revision. RARE is easily incorporated into existing RLHF algorithms and successfully finetunes the pretrianed model. We compare against different reward functions trained on the revision sequences, and we find that RARE has strong quantitative and qualitative results, leading to layouts that are well-aligned, cohesive, and more aligned with human preferences.

**Limitations** RARE faces certain limitations. For example, collecting revisions can be time-consuming, especially for high-dimensional domains like images. Within layout generation works, our work makes certain assumptions, such as the types of available elements. Future work in enabling for incorporating new assets that designers may find relevant may address this.

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

# A APPENDIX

## A.1 HYPERPARAMETERS

### A.1.1 PLAY PRETRAINING

We train a text-conditioned PLay model. We remove the guideline condition and replace it with a text condition, which uses text embedding features from a BERT model with 12 layers, 12 attention heads, and hidden size 768. We inject text conditions through element-wise condition on pooled text embeddings and cross attention with the full text embedding.

The rest of the hyperameters that we used are equivalent to those in Cheng et al. (2023). We train the model on 8 Google Cloud TPU v4 cores for 40,000 steps with batch size 1024.

### A.1.2 REWARD MODEL TRAINING

We include the hyperparameters for training in Table 2.

| Method | Reward Model Pretraining | | Finetuning |
| | CLAY Pretrain Steps | $\mathcal{D}_{\text{human}}$ Train Steps | Optim. Steps |
|---|---|---|---|
| Supervised Finetuning | x | x | 1,000 |
| RARE Keystroke | 2,000 | 200 | 1,000 |
| RARE Preference | 2,000 | 1,00 | 800 |
| RARE Chamfer | 2,000 | 400 | 1,00 |

Table 2: Reward Model Training Hyperparameters.

RARE and the Preference Reward Model have the same architecture as the denoising diffusion model used in PLay, with the exception that there is no time embedding, and there is an additional MLP layer that reshapes the output features and projects it to a scalar prediction. For the Chamfer Reward Model, we reduce the number of layers to 2, number of heads to 4, and key, query and value dimensions to 256 to prevent overfitting.

## A.2 RLHF HYPERPARAMETERS

We train with sample batches of 256. In accordance with DDPO, we compute losses for a single timestep across denoising timesteps together. We set the PPO clip range to 1e-2. We use a batch size of 64 on 8 Google Cloud TPU v4 cores.

## A.3 PLAY COLOR LEGEND

We use the same color legend as in Cheng et al. (2023) to visualize the layouts. Colors for popular class elements are rendered in Figure 8.

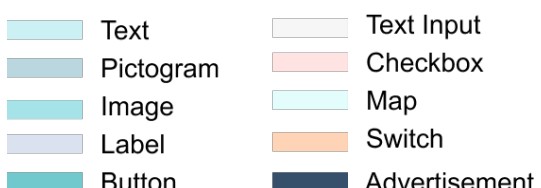

Figure 8: **Visualization Colors**

### A.4 ADDITIONAL DATASET STATISTICS

### A.4.1 CLAY

PLay is trained on CLAY, a large-scale dataset of 59,555 mobile screen layouts. The average number of elements for the original version of CLAY is 19.6, and it contains 24 element classes, including compound class types such as list items and container. To reduce the complexity of editing sequences and increase the number of revised designs we can collect given the limited time and budget, we select 10 classes to simplify the layouts, and the updated mean number of elements per layout is 11.4 (standard deviation 9.0). The distribution of element classes we train on is shown in Figure 9.

### A.4.2 REVISION DATASET

Across the revision dataset, designers make on average 889.3 (standard deviation 612.5) edits. Because the logs are extremely verbose, we condense the sequence of revisions to be every 10 logged steps. Excluding extraneous logs that are not reflected in the PLay layouts (e.g. color or font of an element does not affect the vectorized layout), 89.7% of edits involve rearranging elements and 10.3% involve resizing elements. This is reasonable, as precisely aligning elements and organizing the layout is a more tedious and common part of revision than resizing elements.

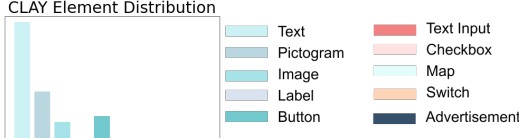

Figure 9: **CLAY Dataset Element Distribution**

In addition, we provide a histogram of the natural log of the number of edits in Figure 10.

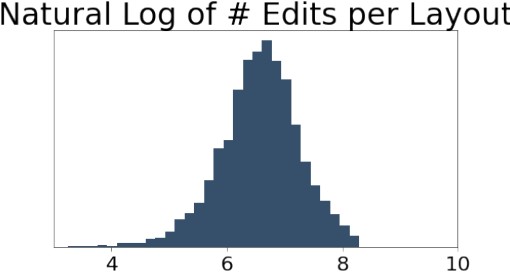

Figure 10: **Distribution of Number of Edits per Layout**

### A.5 ADDITIONAL SAMPLES

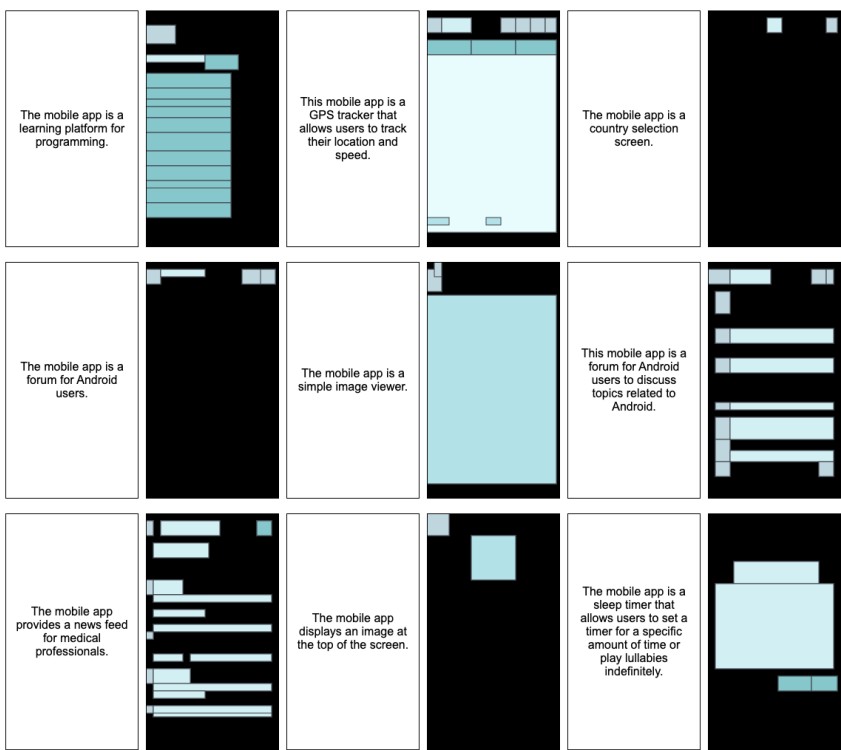

Figure 11: Non-cherrypicked samples from RLHF w/ RARE Keystroke
.

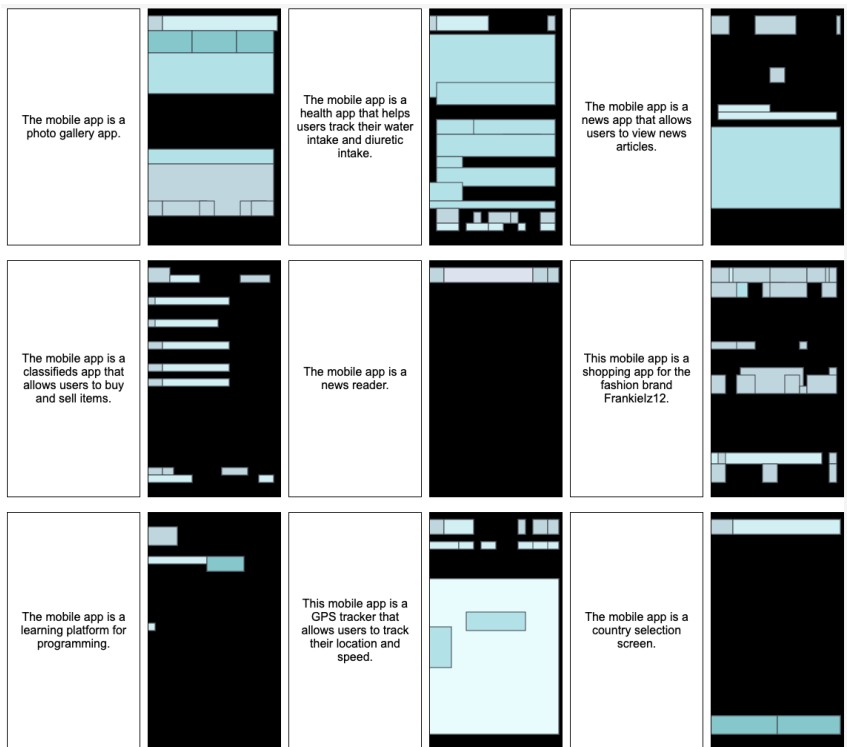

Figure 12: Non-cherrypicked samples from RLHF w/ a preference-based reward model.

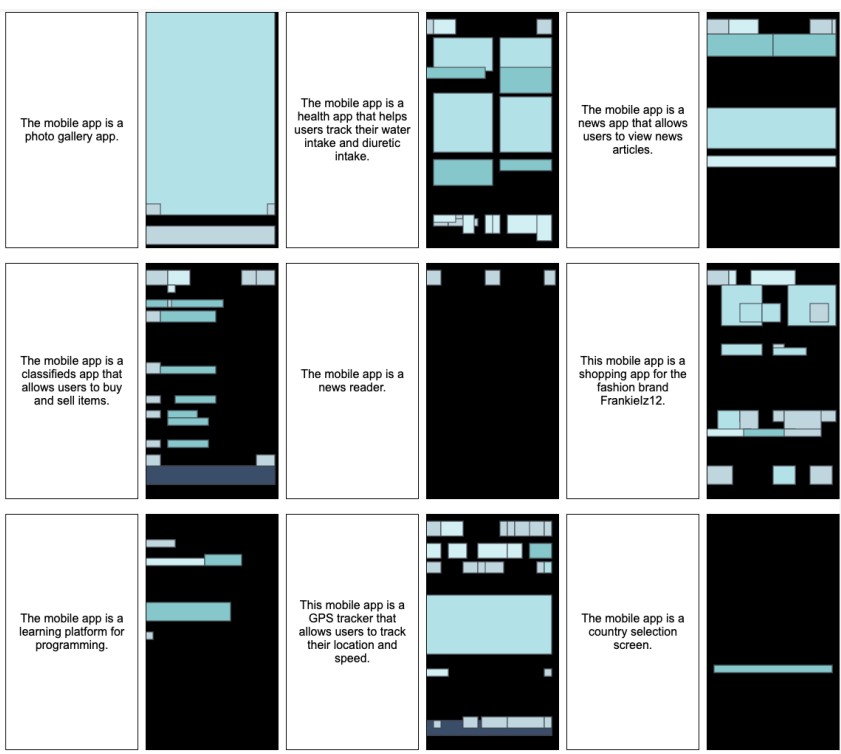

Figure 13: Non-cherrypicked samples from RLHF w/ a RARE Chamfer distance reward model.

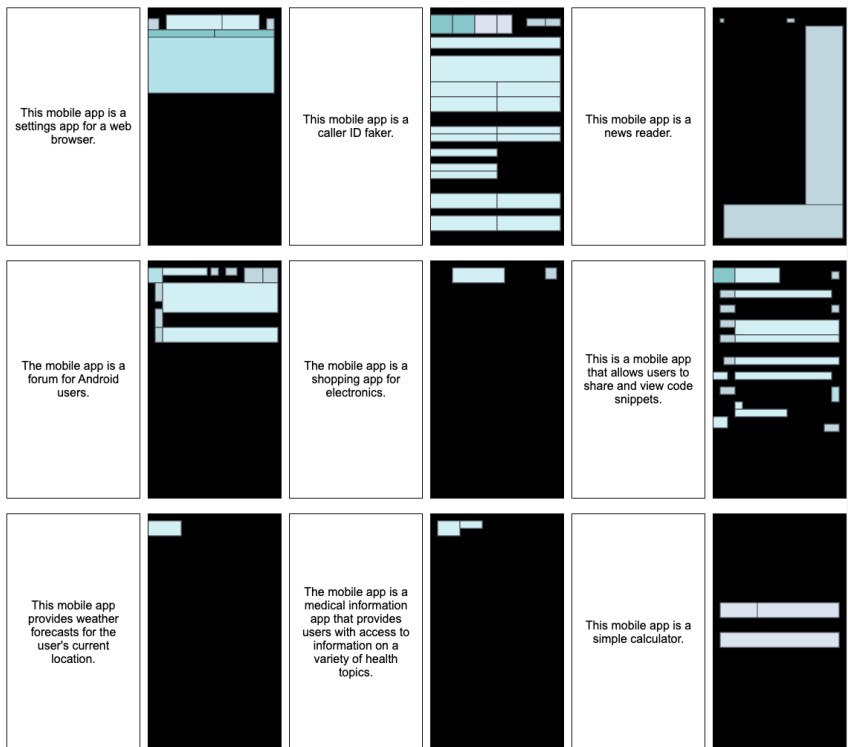

Figure 14: Non-cherrypicked samples from the Supervised Finetuning model.

