# OpenReview forum: "Leveraging Human Revisions for Improving Text-to-Layout Models"
_ICLR.cc/2024/Conference — Submitted to ICLR 2024_

### Official Review · Reviewer_Qjey · 2023-10-26

**Soundness:** 2 fair
**Presentation:** 2 fair
**Contribution:** 2 fair
**Rating:** 6
**Confidence:** 3

**Summary:**

The paper proposes Revision-Aware Reward Models (RARE), which leverages human revisions to strengthen alignment in the context of generative layout models for mobile screens. The authors involve expert designers in fixing layouts generated by a pretrained generative layout model and train a reward model based on how these designers revise the generated layouts. By optimizing the model using reinforcement learning from human feedback (RLHF) with the learned reward model, RARE can enhance the generative model's ability to produce modern, designer-aligned layouts. On a dataset of 276 corrected UI Layouts from designers, the authors compare the proposed method with Supervised Finetuning (SF), Preference Reward + RLHF, Chamfer Distance + RLHF, and the results show the potential of the method quantitatively and qualitatively.

**Strengths:**

1. The proposed method focuses on utilizing nuanced feedback, such as corrections, explanations, and reasoning, to enhance generative models. While prior works have primarily relied on high-level labels, the authors' emphasis on more involved feedback represents a novel perspective. By introducing the concept of Revision-Aware Reward Models (RARE) and applying it to generative layout models, the research offers a unique contribution to the field.

2. The involvement of expert designers in fixing layouts generated by a pretrained model adds credibility to the evaluation process. The training of a reward model based on human revisions and the subsequent optimization of the generative model using reinforcement learning demonstrate a rigorous and systematic approach. The reported results strengthen the overall quality of the work.

3. Overall, the paper is well-written and easy to follow. It provides a clear explanation of the research objectives and the proposed methodology. The authors effectively communicate the significance of utilizing nuanced feedback and human revisions in improving generative models.

**Weaknesses:**

1. The analysis of the dataset used in this work is not comprehensive. It would benefit the research to provide an overview of the dataset, including its characteristics and why the proposed method is well-suited for this specific dataset. Additionally, to ensure the generalizability and robustness of the proposed approach, it is crucial to evaluate its performance on diverse datasets or domains. Conducting experiments with the RARE approach on different types of layouts, such as web or desktop interfaces, would provide a more comprehensive understanding of its effectiveness and applicability in various contexts.

2. While human revisions are utilized to train the reward model, there is a notable absence of analysis or insights into the specific patterns, reasoning, or design principles underlying these revisions. Gaining a deeper understanding of the factors driving the revisions made by human designers would offer valuable insights for further improving the generative model. Conducting a thorough analysis of the revisions, such as identifying common patterns or design choices, would enrich the understanding of the alignment between human values and generative outputs and provide guidance for refining the model's performance.

3. The qualitative results presented in the paper lack detailed analysis, as only a few examples are provided. To strengthen the research, it would be beneficial to offer expert opinions and insights to support and explain why the results obtained with the RARE approach are considered good.

4. Some minor typos are present in the paper, such as in section 5.2, paragraph 2, where the first sentence contains two instances of the word "that."

**Questions:**

1. What is the size of the collected dataset?
2. For the qualitative results, are there any implications derived?
3. As the data collection involves human subjects, is the study proved by IRB?

---

> ### Author Response · Authors · 2023-11-23
>
> Thank you very much for your detailed comments and constructive feedback for improving the paper. To address your concerns, we have revised the paper thoroughly, including further data collection, new analysis, and additional evaluation metrics. We respond to each of your questions here and have made these improvements in the revised manuscript.
>
> **New Data**
>
> We have collected more data from raters trained specifically on this task. We provided instructions and worked with them through a few iterations to ensure strong data quality and consistency. Through this process, we have collected **836** new data examples, around **4x** our initial dataset of 276 revision. Our design revision dataset, which will consist of **over 3,000 revisions**, will be released to the public on the GitHub.
>
> We have included results with supervised finetuning, RARE Keystroke, RARE Chamfer Distance, and RARE Preference. Our evaluation metrics are FID score and DocSim. For our FID score calculations, we calculate FID statistics for a held-out batch of 128 samples. We take 1024 samples and calculate the FID statistics in batches of 128, to match the evaluation batch size, and report the average of the FID scores. The DocSim metric is a standard way of evaluation document similarity, proposed in [1], which we include for an additional evaluation metric.
>
> In our new results across 3 seeds, we find that RARE Chamfer achieves the lowest FID score. In addition, RARE methods achieve higher DocSim scores than supervised finetuning. We include updates samples from RARE and baselines in the revised copy of our paper. Our results shows that revision-aware reward (RARE) methods are successful in better aligning the pretrained layout model with human revisions.
>
> [1] https://arxiv.org/pdf/1909.00302.pdf
>
> **Updated Results**
>
> |                       | FID Score   | DocSim        |
> |-----------------------|-------------|---------------|
> | CLAY Dataset          | 73.4 +- 4.5 | 0.14 +- 0.01  |
> | Revision Dataset      | 63.1 +- 3.0 | 0.16 +- 0.01  |
> | PLay Samples          | 76.6 +- 0.3 | 0.31 +- 0.002 |
> | Supervised Finetuning | 68.9 +- 0.6 | 0.26 +- 0.01  |
> | RARE Keystroke        | 70.1 +- 0.8 | 0.27 +- 0.03  |
> | RARE Preference       | 72.4 +- 2.6 | 0.31 +- 0.02  |
> | RARE Chamfer          | 68.8 +- 0.5 | 0.28 +- 0.01  |

---

> > ### Author Response · Authors · 2023-11-23
> >
> > **1. What is the size of the collected dataset?**
> >
> > Please see the above.
> >
> > **2. For the qualitative results, are there any implications derived?**
> >
> > We plan to evaluate our results with human designers in the near future to gain more insight on this. We have just recently wrapped up our data collection process, where we have increased our dataset to 3,000 revisions, which is orders of magnitude larger than our initial dataset of 276 revisions. We hope to have this very soon!
> >
> > **3. As the data collection involves human subjects, is the study proved by IRB?**
> >
> > Yes.
> >
> > **4. Analysis of dataset**
> >
> > Thank you for this comment. We have provided more details on the CLAY dataset in the Appendix. The average number of elements for the original version of CLAY is 19.6, and it contains 24 element classes, including compound class types such as list items and container. We include a plot of the element distribution in the Appendix.
> >
> > Unfortunately, due to time and financial constraints, we are unable to conduct this on more extensive domains. Collecting human revisions is time consuming and expensive, and we seek to show that this style of data collection can lead to successful alignment to human values in our specific domain, in hopes and expectations that this method or motivation can be applied to other domains. Particularly for web or desktop interfaces, we would expect our method to have a similar performance, as the overall mockup style would be similar, with the main difference being the human alignment captured through revisions.
> >
> > **5. Thorough analysis of revisions**
> >
> > Thank you for this question. We have included some analysis of the types of edits made in the paper:
> >
> > Across the revision dataset, designers make on average 889.3 (standard deviation 612.5) edits. Because the logs are extremely verbose, we condense the sequence of revisions to be every 10 logged steps. Excluding extraneous logs that are not reflected in the PLay layouts (e.g. color or font of an element does not affect the vectorized layout), 89.7% of edits involve rearranging elements and 10.3\% involve resizing elements. This is reasonable, as precisely aligning elements and organizing the layout is a more tedious and common part of revision than resizing elements.
> >
> > Qualitatively, we are able to replay and rewatch the revision process. We notice that humans tend to take a coarse-to-fine approach to revising, by first roughly moving elements and then refining the layout to be better aligned. We have also noticed that the designers commonly copy and paste structured collections of elements: for instance, the designer may first pair text and image together and then duplicate it many times in a list-like fashion. We hope that these types of design habits are implicitly captured in the reward models, which are trained on these edit sequences. We do not do any explicit modeling or hard-coding of reward functions based on observations.
> >
> > **6. Typos**
> >
> > Thank you for pointing out the typos. We have revised them and clarified equations in an updated draft of our paper.

---

### Official Review · Reviewer_d2Zb · 2023-10-31

**Soundness:** 3 good
**Presentation:** 3 good
**Contribution:** 2 fair
**Rating:** 5
**Confidence:** 3

**Summary:**

The paper collects a dataset of sequences of human designers revising model-generated mobile app layouts. The paper proposes a method called RARE to learn a reward model based on the collected data and use the reward model to perform RLHF finetuning on a pre-trained layout generation model. Experimental results show that the proposed method is better than simple finetuning or other reward models.

**Strengths:**

- The paper collects high-quality datasets from expert app layout designers. The dataset could be important for the community.
- The method is simple but effective.
- The proposed method is much more effective than simple finetuning.

**Weaknesses:**

- The novelty might be limited. It seems that the novel part of the method is how training samples are constructed from the collected dataset to train RARE. Other parts like the diffusion models and RLHF are similar to existing work.
- I am not quite convinced by sec. 4.2 on the reward model pretraining. The construction of pretraining data seems a bit too heuristic and are not grounded on any reasonable arguments/observations. Why do you assume dropping needs 1 time step, revised elements needs 2 time step, and added element needs 3 time steps? The parameters for each operation, e.g. resize 0.5-2 times have no explanation as well.
- Limited evaluation metrics. Previous work has considered other metrics like NLL, Coverage, or Overlap. Are these reasonable metrics for the experiments considered in this paper? If not, are there other possible metrics beyond FID? Is it reasonable to conduct user study to make the results more convincing?

**Questions:**

Table 1 implies that RARE and Preference has the same FID, yet Fig. 6 shows that RARE is better than Preference. Is it the case that RARE is better than Preference for most of the evaluation samples? If so, why do these methods have the same FID?

---

> ### Author Response · Authors · 2023-11-23
>
> Thank you very much for your detailed comments and constructive feedback for improving the paper. To address your concerns, we have revised the paper thoroughly, including further data collection, new analysis, and additional evaluation metrics. We respond to each of your questions here and have made these improvements in the revised manuscript.
>
> **New Data**
>
> We have collected more data from raters trained specifically on this task. We provided instructions and worked with them through a few iterations to ensure strong data quality and consistency. Through this process, we have collected **836** new data examples, around **4x** our initial dataset of 276 revision. Our design revision dataset, which will consist of **over 3,000 revisions**, will be released to the public on the GitHub.
>
> We have included results with supervised finetuning, RARE Keystroke, RARE Chamfer Distance, and RARE Preference. Our evaluation metrics are FID score and DocSim. For our FID score calculations, we calculate FID statistics for a held-out batch of 128 samples. We take 1024 samples and calculate the FID statistics in batches of 128, to match the evaluation batch size, and report the average of the FID scores. The DocSim metric is a standard way of evaluation document similarity, proposed in [1], which we include for an additional evaluation metric.
>
> In our new results across 3 seeds, we find that RARE Chamfer achieves the lowest FID score. In addition, RARE methods achieve higher DocSim scores than supervised finetuning. We include updates samples from RARE and baselines in the revised copy of our paper. Our results shows that revision-aware reward (RARE) methods are successful in better aligning the pretrained layout model with human revisions.
>
> [1] https://arxiv.org/pdf/1909.00302.pdf
>
> **Updated Results**
>
> |                       | FID Score   | DocSim        |
> |-----------------------|-------------|---------------|
> | CLAY Dataset          | 73.4 +- 4.5 | 0.14 +- 0.01  |
> | Revision Dataset      | 63.1 +- 3.0 | 0.16 +- 0.01  |
> | PLay Samples          | 76.6 +- 0.3 | 0.31 +- 0.002 |
> | Supervised Finetuning | 68.9 +- 0.6 | 0.26 +- 0.01  |
> | RARE Keystroke        | 70.1 +- 0.8 | 0.27 +- 0.03  |
> | RARE Preference       | 72.4 +- 2.6 | 0.31 +- 0.02  |
> | RARE Chamfer          | 68.8 +- 0.5 | 0.28 +- 0.01  |
>
> **1. RARE vs. Preference FID**
>
> Please see the above
>
> **2. Novelty of our method**
>
> Thank you for this comment. We agree that the overarching framework of our method (diffusion models and RLHF) are similar to prior work. However, the motivation of our method is to enhance the reward model used within RLHF to take in stronger forms of feedback, such as human corrections. In addition, to our knowledge, this is first work using human feedback to improve generative layout models.
>
> **3. Reward Model Pretraining**
>
> Thank you for this comment. We select these parameters heuristically, based on initial analysis of the edited dataset. For instance, removing an element is logged as 1 step. Further investigation of how best to generate pretraining data for the reward model can be useful, though we leave this to future work. We chose to pretrain the reward model to avoid overfitting to our the revision dataset.
>
> **4. Additional evaluation metrics**
>
> Please see the above. We have included a new DocSim metric.

---

### Official Review · Reviewer_otPj · 2023-11-01

**Soundness:** 2 fair
**Presentation:** 2 fair
**Contribution:** 3 good
**Rating:** 5
**Confidence:** 4

**Summary:**

The paper proposes improving the layout generator model with human revision feedback. The work experiments with using the time-taken by human editor, distance between the revision, sft, and binary preferences to test improving the model RLHF.

**Strengths:**

The paper investigates using more nuanced feedback rather than binary preference, which is a less explored area of research.

**Weaknesses:**

The papers outline is not in a typical format, the dataset description comes later, I struggled to imaging the dataset while reading the experiments section without reading the dataset description before.

The equations used in the paper aren't fully explained and in some places the symbols used in the equation and the description are inconsistent. I did not get a full understanding of the background reading the paper because of this. Maybe the authors can reduce the size of the figures or move them to the appendix section to get more space.

The experiment aren't rigorous and the results were not analyzed properly
- The explanation of why the Chamfer distance did worse than even the preference-based model isn't convincing. The appendix section shows that Chamfer models were trained by much more iterations in all the steps (49000 vs 2000) than all the other models, could it be just that the Chamfer model just got overfitted? To analyze the problem the author should compare chamfer vs time distance distributions and/or have more comparable training iteration numbers to start with.
- Since the proposed approach performs similarly to the preference-based model, the authors should investigate more into this with more random seeds to start with.

**Questions:**

Although this is a more recent work, it might be worth looking into this work which also looks into using human revision information https://arxiv.org/abs/2310.05857.

Can you train without RL? Like what is done in the paper? It probably needs alignment between the human edits (which element got changed into what).

I am surprised that there were no guidelines for humans who revised the layout. Have you seen any undesirable edits in the dataset?

Although it isn't completely clear, I am assuming you are using every edit by a person on the layout as a separate edit. It might be worth clustering some of them.

According to the description, reward model predicts the time/distance between the revisions. Does it then mean that it is a penalty model rather than a reward model?

---

> ### Author Response · Authors · 2023-11-23
>
> Thank you very much for your detailed comments and constructive feedback for improving the paper. To address your concerns, we have revised the paper thoroughly, including further data collection, new analysis, and additional evaluation metrics. We respond to each of your questions here and have made these improvements in the revised manuscript.
>
> **New Data**
>
> We have collected more data from raters trained specifically on this task. We provided instructions and worked with them through a few iterations to ensure strong data quality and consistency. Through this process, we have collected **836** new data examples, around **4x** our initial dataset of 276 revision. Our design revision dataset, which will consist of **over 3,000 revisions**, will be released to the public on the GitHub.
>
> We have included results with supervised finetuning, RARE Keystroke, RARE Chamfer Distance, and RARE Preference. Our evaluation metrics are FID score and DocSim. For our FID score calculations, we calculate FID statistics for a held-out batch of 128 samples. We take 1024 samples and calculate the FID statistics in batches of 128, to match the evaluation batch size, and report the average of the FID scores. The DocSim metric is a standard way of evaluation document similarity, proposed in [1], which we include for an additional evaluation metric.
>
> In our new results across 3 seeds, we find that RARE Chamfer achieves the lowest FID score. In addition, RARE methods achieve higher DocSim scores than supervised finetuning. We include updates samples from RARE and baselines in the revised copy of our paper. Our results shows that revision-aware reward (RARE) methods are successful in better aligning the pretrained layout model with human revisions.
>
> [1] https://arxiv.org/pdf/1909.00302.pdf
>
> **Updated Results**
>
> |                       | FID Score   | DocSim        |
> |-----------------------|-------------|---------------|
> | CLAY Dataset          | 73.4 +- 4.5 | 0.14 +- 0.01  |
> | Revision Dataset      | 63.1 +- 3.0 | 0.16 +- 0.01  |
> | PLay Samples          | 76.6 +- 0.3 | 0.31 +- 0.002 |
> | Supervised Finetuning | 68.9 +- 0.6 | 0.26 +- 0.01  |
> | RARE Keystroke        | 70.1 +- 0.8 | 0.27 +- 0.03  |
> | RARE Preference       | 72.4 +- 2.6 | 0.31 +- 0.02  |
> | RARE Chamfer          | 68.8 +- 0.5 | 0.28 +- 0.01  |

---

> > ### Author Response · Authors · 2023-11-23
> >
> > **1. Comparison to Yao et. al.**
> >
> > Thanks for the reference! This work also uses human revisions to align large models (in their case, a large language model) with human objectives. As far as we can tell, this work was released after the ICLR deadline and is concurrent work. The main difference is that Yao et. al. uses a supervised objective (roughly comparable to Direct Policy Optimization), whereas we use RLHF with an improved reward model.
> >
> > **2. Training without RLHF**
> >
> > Yes, Yao et. al. should be adaptable to our project domain. However, one difference in our work is that we are able to leverage sequential data, i.e. the entire edit sequence generated by the human revisor. On the other hand, Yao et. al. proposes using only the AI and edited output.
> >
> > **3. Guidelines for Human raters**
> >
> > Thank you for this question. To clarify, we use “guidelines” in multiple senses here: (1) **Guidelines (guides, grids, lines)** are a type of condition for the layout generation, proposed in PLay [1]. (2) **Guidelines are instructions for the raters.**
> >
> > We do not use guideline/grid conditions for PLay, and we instead use text conditions. However, we do provide guidelines/instructions to the raters.
> >
> > For our first round of data collection, we provided basic guidelines to the designer, but we did not restrict designers to specific design rules, in order to generate a diverse dataset generated by experienced designers.
> >
> > We have conducted an additional round of data collection from 4 trained raters. For our new round of data collection, we have provided more detailed instructions and worked closely with raters to ensure that they understand the task and plugin. Our onboarding process consisted of providing raters with examples of desired edits and critiquing and assessing a few rounds of their revisions.
> >
> > [1] https://arxiv.org/pdf/2301.11529.pdf
> >
> > **4. Post-processing human edits**
> >
> > Thank you for this comment. We do additional post-processing of edits, where we cluster edits in groups of 10. We use the Figma API to log any edits in the layout. We have added this information to the appendix.
> >
> > **5. Penalty vs. Reward Model**
> >
> > Yes, our model can also be interpreted as a penalty model. We used reward instead of penalty following previous RLHF works. In the general literature of reinforcement learning, cost/penalty and reward is often used interchangeably (https://rail.eecs.berkeley.edu/deeprlcourse/deeprlcourse/static/slides/lec-2.pdf).
> >
> > **6. The paper’s outline is not in a typical format. The dataset description comes later.**
> >
> > Thank you for this comment. A brief overview of the datasets provided is the first paragraph of the experiments section Section 5.1 in our initial draft, and we further describe the human correction dataset in the last paragraph of Section 5.1. However, we agree that the dataset description is very useful to the reader, and we have provided additional information in the experiments section and appendix.
> >
> > **7. Equations not fully explained**
> >
> > Thank you for pointing this out. We have fixed some inconsistencies and improved clarity in the revised manuscript.
> >
> > **8. Experiments: Comparing Chamfer vs. Time Distance Distributions**
> >
> > Please see the above. We include updated hyperparmeters in the paper, where Chamfer is trained with a similar number of steps.
> >
> > **9. Experiments: Running on more random seeds**
> >
> > Thank you for this comment. We have updated our results with 3 seeds!

---

### Official Review · Reviewer_RQ9B · 2023-11-06

**Soundness:** 2 fair
**Presentation:** 2 fair
**Contribution:** 2 fair
**Rating:** 5
**Confidence:** 3

**Summary:**

This paper proposes to train a reward model from human designer revisions on generated layouts by a pre-trained layout model. Then, they optimizer the down-stream model by RLHF with the trained reward model. In this way, RARE aligns the model with human preference to produce more designer-aligned layout.

**Strengths:**

1. This paper proposes a novel approach to integrate different human feedbacks into model training, i.e., the step-by-step revision sequences.
2. The reward is designed to correlate with revision time, which provides better signals than binary comparison rewards.

**Weaknesses:**

1. Though the paper presents a new notion of human feedback, i.e., revision sequences, its application to layout generation makes its applicability quite constrained. The first time I read the abstract, I thought the paper seemed to propose a general methodology for RLHF. After going through the paper, I realized that the proposed reward training is only specifically designed for the text-to-layout generation domain.
2.	The evaluation is not sound to me. In section, the major quantitative evaluation results are presented in Table 1; the remaining evaluation are mainly shown by generative examples. All of results are about one task. More quantitative evaluation evidence can make the conclusion sounder.
3.	The presentation can be improved. The equation (4) can be confusing. It will be better to list them following time ordering.

**Questions:**

Can authors provide more details of the CLAY dataset, like its statistics?

---

> ### Author Response · Authors · 2023-11-23
> **Response to Reviewer RQ9B**
>
> Thank you very much for your detailed comments and constructive feedback for improving the paper. To address your concerns, we have revised the paper thoroughly, including further data collection, new analysis, and additional evaluation metrics. We respond to each of your questions here and have made these improvements in the revised manuscript.
>
> **New Data**
>
> We have collected more data from raters trained specifically on this task. We provided instructions and worked with them through a few iterations to ensure strong data quality and consistency. Through this process, we have collected **836** new data examples, around **4x** our initial dataset of 276 revision. Our design revision dataset, which will consist of **over 3,000 revisions**, will be released to the public on the GitHub.
>
> We have included results with supervised finetuning, RARE Keystroke, RARE Chamfer Distance, and RARE Preference. Our evaluation metrics are FID score and DocSim. For our FID score calculations, we calculate FID statistics for a held-out batch of 128 samples. We take 1024 samples and calculate the FID statistics in batches of 128, to match the evaluation batch size, and report the average of the FID scores. The DocSim metric is a standard way of evaluation document similarity, proposed in [1], which we include for an additional evaluation metric.
>
> In our new results, we find that RARE Chamfer achieves the lowest FID score. In addition, RARE methods achieve higher DocSim scores than supervised finetuning. We include updates samples from RARE and baselines in the revised copy of our paper. Our results shows that revision-aware reward (RARE) methods are successful in better aligning the pretrained layout model with human revisions.
>
> [1] https://arxiv.org/pdf/1909.00302.pdf
>
> **Updated Results**
>
> |                       | FID Score   | DocSim        |
> |-----------------------|-------------|---------------|
> | CLAY Dataset          | 73.4 +- 4.5 | 0.14 +- 0.01  |
> | Revision Dataset      | 63.1 +- 3.0 | 0.16 +- 0.01  |
> | PLay Samples          | 76.6 +- 0.3 | 0.31 +- 0.002 |
> | Supervised Finetuning | 68.9 +- 0.6 | 0.26 +- 0.01  |
> | RARE Keystroke        | 70.1 +- 0.8 | 0.27 +- 0.03  |
> | RARE Preference       | 72.4 +- 2.6 | 0.31 +- 0.02  |
> | RARE Chamfer          | 68.8 +- 0.5 | 0.28 +- 0.01  |
>
> **1. Can the authors provide more details of the CLAY dataset, like its statistics?**
>
> Thanks for this comment. We have added some additional information about the CLAY dataset in the appendix.
>
> **2. General methodology for RLHF**
>
> Thank you for this comment. We focus specifically on text-to-layouts, which we describe in the abstract and title, but we will make this more clear in the abstract.
>
> Our work has been inspired by the general progress in the RLHF community, specifically in the works exploring different forms of human feedback. Although our method has only been applied to UI layouts and may not scale to other domains, especially high-dimensional ones such as images, we hope that training reward models on human data other than human preferences can yield promising results for RLHF. We will make clear in the revised paper that our reward model is specifically designed for the text-to-layout domain.
>
> **3. Evaluation Results**
>
> Please see the above.
>
> **4. Presentation**
>
> Thank you for your comments. Equation 4 was modified from the DDPO paper, but we have updated the ordering of variables to be clearer.

---

### Meta-Review · Area_Chair_mzAf · 2023-12-06

**Metareview:**

The paper addresses the improvement of layout generator models using human revision feedback. To achieve this, it proposes a method called RARE (Revision-Aware Reward Models) for learning a reward model and conducts RLHF fine-tuning on a pre-trained layout generation model. Experiments demonstrate that the proposed method outperforms established fine-tuning approaches and other reward models. The reviewers have highlighted several strengths of this work, including:
1. The paper explores the use of nuanced feedback rather than binary preferences, which is a less explored area of research. Correlating the reward with measures such as revision time appears to provide more informative signals compared to binary comparison rewards.
2. The paper gathers high-quality data from expert app layout designers, which could be valuable to the community.
3. The involvement of expert designers in correcting layouts generated by a pre-trained model lends credibility to the authors' evaluation.

However, after considering the authors' responses, the reviewers have several reservations about accepting this paper due to the following weaknesses:

1. Concerns about limited novelty: While the work is well-motivated, as suggested in Strength 1, the level of novelty involved is not particularly significant. It appears that the reward model extends from binary to continuous values representing revision times, and the reward learning phase is not a challenging problem (at least not in the setting of this work). The rest of the work (diffusion model and RLHF) is familiar in the existing literature.
2. Confusion with the many changes in the paper and rebuttal: Some of the reviewers were perplexed by the updated paper, as it didn’t clearly highlight what was changed, and they were confused by several points, e.g., (1) Chamfer performed worst in the paper because of overfitting, and it performed the best in the new results. (2) There are also some inconsistencies comparing the paper and the rebuttal on the post-processing technique used (e.g., clustering vs. raw). In light of these inconsistencies and many changes, one reviewer suggested it would be best to rewrite the paper and have it reviewed with a fresh set of eyes.
3. Presentation is sometimes quite problematic: Poor background explanation, dataset description, experiment setup, and results analysis. While this work appears to be the first to use RLHF with fine-grained feedback for layout generation, many details of the methodology design lack proper explanation or justification and could seem arbitrary.
4. The paper has a relatively narrow scope for a conference such as ICLR, as it focuses on generative layout models for mobile screens.

The inclination of three out of the four reviewers is to reject this paper. I find that the weaknesses noted above outweigh the aforementioned strengths, so I see no reason to reverse the majority opinion and, therefore, recommend rejecting this paper.

**Justification For Why Not Higher Score:**

The reviewers generally felt that the paper needs some extensive rewrite before being resubmitted (see weaknesses 2 and 3).

**Justification For Why Not Lower Score:**

N/A

---

### Decision · Program_Chairs · 2024-01-16

Reject